# Integrated Systems of a Solar Thermal Energy Driven Power Plant

**Yasser Abbas Hammady AL-Elanjawy** [1,2] **and Mustafa Yilmaz** [2,*]

1   Ministry of Electricity CEP/Middle Region Baghdad, University of Technology, Baghdad 10066, Iraq; yasserhammady@marun.edu.tr
2   Institute of Pure and Applied Sciences, Marmara University, 34722 Istanbul, Turkey
*   Correspondence: mustafa.yilmaz@marmara.edu.tr

**Abstract:** As a consequence of the limited availability of fossil fuels, green energy is gaining more and more popularity. Home and business electricity is currently limited to solar thermal energy. Essential receivers in current solar thermal power plants can endure high temperatures. This ensures funding for green thermal power generation. Regular solar thermal power plant testing is arduous and time-consuming. They need expensive installation and take up much space. Many free software and tools can model and simulate solar thermal-producing systems. Some techniques can evaluate and predict the plant's performance, while others can investigate specific components. Nothing using research tools has ever reached the top. Simulated testing may precede power plant construction. This research requires basic visual help. A rudimentary plant model was developed when the computational calculations for thermal performance were obtained. Plus, it may estimate how much power the facility would produce. The program includes hydraulic heat transport fluids, ambient factors, a database, and user input parameters. Free hourly weather data from anywhere is available from the simulator. The simulator calculates the thermal power delivered by each component while running constituent simulators.

**Keywords:** energy; solar thermal; power plant

## 1. Introduction

Solar power is one of the many sources of renewable energy that are now available. The use of concrete solar power (CSP) and photovoltaic technologies are two methods that may be used to harvest solar energy. It is possible that the thermal energy reserves of the system will be used in the event that there is insufficient solar power to operate the turbine. Because of this, solar thermal power plants are better than photovoltaics in terms of efficiency. Even though they are quite strong, the sun's beams are not powerful enough to create energy on Earth. Large mirrors are used to focus the sun's rays that are on their way to the absorbers before they reach them. An absorber is a fluid that transports heat and absorbs radiation heat [1]. Radiation heat is absorbed this way. The turbine is driven by the thermal energy of the fluid, which ultimately results in the production of electricity. When it comes to the generation of energy, solar thermal power plants often make use of the central receiver and the parabolic trough designs.

Essentially, the technique doubles as a storage medium for the thermal energy collected from the sun. Two tanks store the fluid; one is heated to a high temperature, and the other is maintained at a low temperature. The solar collector or receiver's job is to raise the fluid's temperature from the low-temperature tank. The fluid is moved to the high-temperature tank for storage once it has been heated to the required temperature using solar energy. Using a heat exchanger, the fluid from the high-temperature tank is transformed into steam, which is then utilized to generate power. Regarding sophisticated hybrid cycles, General Electric calculated the heat balance. Their research showed that a hybridization method combining evaporation and superheating was the most effective for their purposes [2].

By including a thermal storage system (TSS), a computer model was developed that encompasses the economics and energy flows inside a solar-fossil fuel hybrid power plant. Based on the data collected by the model, we can see how much electricity was drawn from the TSS, how much went to which load, how much went back to the TSS, and how much fuel the fossil section of the hybrid power plant consumed. The model determined the power produced by an SCR hourly or quarter-hourly. They proved that for a solar-fossil fuel hybrid system to be financially feasible, the capital investment could not be more than 2.5 times the asset's present value for a fossil fuel power system comparable to the hybridization. Modeling, simulating, and evaluating such systems was the primary aim of this research. Solar central receivers (SCRs), parabolic trough collectors (PTCs), compact linear Fresnel collectors (CLFCs), and solar dishes (SDs) were the focus of this research. These definitions are provided here to compare different hybrid management systems. The hybridization of redundant systems, parallel fossil heaters, solar-enhanced systems, and solar preheat systems were the four hybridization options that were compared. It was proposed that well-built hybrid plants would provide considerable advantages over solar plants that rely only on solar energy, particularly for upcoming markets. By making the most of these possibilities, we may obtain cheaper energy costs, more incredible value energy due to dispatchability, less capital investment in new technologies, and enhanced energy conversion efficiency [3].

The present situation has resulted in an exponential increase in the need for energy across all industries, including production, infrastructure, etc. Only 30–34% of the world's total energy usage goes toward meeting the building's heating and cooling needs. Their actions have given birth to several health issues, such as pollution control, warming the planet, a diminished ozone layer, and so on. The governance of energy and security has recently captured the global spotlight [4]. According to the Intergovernmental Panel on Climate Change (IPCC), greenhouse gases are largely generated by climate change and global warming. It has been suggested by academics that individuals may lessen their impact on the environment by using renewable energy sources. Various renewable possibilities are accessible. Their selection should be based on meeting various criteria such as techno-economic, environmental concerns, geographical conditions, desired energy quality, etc. The energy intensity of a country and its availability around the clock are now the most important benchmarks for comparing nations. Energy intensity measures how much power is used in relation to the GDP. The benefit of this is often greater in poorer nations than in more advanced ones. A greater value indicates a substantial reliance on energy. India utilizes around 6% of the world's primary energy [5]. Section 2 of this article reviews the literature, Section 3 explains the research methods, Section 4 discusses this study's results and debate, and Section 5 offers a summary and suggestions for further study.

## 2. Related Work

This article explains the most recent assessments on solar thermal energy-driven power plants' integrated systems. There was strong experimental and modeled agreement in Almeria, Spain, using a hybrid CSP that used two TESs [6]. Molten salt (MS) pumping and free drainage between the hot and cold tanks were the thermal energy storage technologies.

A hybrid solar-powered system is suggested in [7] for industrial sustainability. An economical, low-concentration, high-optical-efficiency parabolic trough collector with a custom-built bio-driven boiler to smooth out solar power variations is the subject of this comprehensive optimization and techno-economic-environmental study.

A decarbonization approach for power plants, which is adaptable and sustainable, is introduced in [8]. The device incorporates a solar energy-powered hybrid membrane-amine carbon capture system. A 19.4% increase in power production and a 10.3% decrease in carbon intensity are the results of the design. There is a 6.9% decrease in specific reboiler duty and a 44% decrease in packing volume.

The 8 MWh[th] prototype plant for molten salt production was conceived, built, and tested [9]. The heat exchanger and storage tanks, two key components, were extensively tested. Big commercial CSP initiatives utilizing an MS TES yielded satisfactory results.

Compared to sensible and latent heat storage, thermo-chemical storage (TCS) systems were determined to need a greater amount of study in [10]. According to the analysis, there is a good chance that solar power may fulfill the world's energy demands (as briefly described in [11]). At the moment, there is not much room for role-playing. Along with the most recent concentrators, you will find details on how to assess various factors and choose the most economical choice. Concentrating solar technology as it is now was analyzed in [12]. The optical and thermal qualities of the concentrate, together with its design, manufacturing, modification, testing, monitoring system, and materials, were covered. According to what is stated in [13], this may be achieved while staying under the thermodynamic concentration limit. A parabolic solar energy system was created in [14] with the help of such instructions and recommendations. Solar power, food processing, and other fields have recently made strides. It was expected that the benefits of commercializing and implementing the authorized technology would materialize. In addition, feedback from actual users was solicited for this technology. According to the author, optimizing the architecture of a solar tubular receiver is crucial for improving its operating efficiency [15]. We looked at the problematic condition that was affecting its performance.

The volumetric receiver design was thoroughly examined in [16] in an effort to minimize the amount of heat lost. Volumetric and tube receivers, which operate in various ways and have distinct geometries, were compared.

According to [17], In addition to generating electricity, the suggested system generates cooling, fresh water, domestic hot water, and hydrogen. Energy and exergy efficiencies are used to evaluate the system's overall performance based on thermodynamic analysis. Parametric studies look at how changing several operational factors and circumstances affects performance. At various stages of the process, the cost rate is estimated.

A novel thermal system called MiniStor is introduced in the article [18]. It employs a thermochemical heat storage (TCM) method based on a reversible reaction between an ammoniated calcium chloride salt and the ammonia ($CaCl_2/NH_3$) cycle to produce heat and coolness. An integrated thermal system featuring photovoltaic thermal collectors, flat plate solar collectors, a thermal conductor module (TCM), and phase change material (PCM) units for energy storage was modeled in Aspen Plus Dynamics using Matlab/Simulink.

Thermodynamic analysis and a novel integrated solar-energy-based system for producing both power and potable water are presented in [19,20]. Solar towers with volumetric solar receivers, solar-powered Rankine cycles, subsystems for molten salt storage and multistage flash distillation (MFD), and other components comprise the suggested system. Every part of the system has its total energy and exergy efficiency determined. Also computed are the proposed system's power generating and freshwater production capacities.

Solar and geothermal energy incorporated into an intergenerational system is the way to go for a product strategy that spans generations [21]. The system's many components are components for drying, thermal energy storage, absorption chilling, space heating heat pumps [22], two ORC power turbines, and more. Thermodynamic studies of the current system and system and subsystem performance evaluations are carried out using energy and exergy methods in four scenarios: single-generation, cogeneration, tri-generation, and multigeneration.

A system-driven design of flexible nuclear technology is successfully proposed in [23], which uses a system modeling technique to find configurations of flexible nuclear reactors that minimize investment and operating costs in a decarbonized energy system. The article presents case studies investigating how system elements affect the decisions made about plant layout. According to the findings, cost-effective, flexible nuclear setups should change depending on their surroundings.

The possibility of solar energy—heat and electricity—significantly contributing to lowering $CO_2$ emissions from industrial heating needs is explored in the research of [24].

To find the best configuration for both systems, the article compares a hybrid system to a traditional system that uses solely natural gas and looks at how much money and pollution it saves. While the basic gas-only system increased costs by 75%, the optimized compact parabolic trough system cut $CO_2$ emissions by 45% in hot climates.

Solar heat's role in decarbonization is explored in a Nordic district heating (DH) network that derives most of its heat from biofuels [25]. Our heating system model was created using Apros® simulation software and data from a Finnish town. We evaluate many decarbonization scenarios for the present heating system, considering elements like solar thermal collector number and design, thermal energy storage (TES), and solar heat constraints.

A power system model (Dispa-SET) is utilized to analyze this coupling pathway's operational costs, efficiencies, and $CO_2$ emissions [26]. Thermal power plants will still exist, and district heating networks may help decarbonize the built environment by using their surplus heat. The analysis is performed for the existing and future European power systems. Thermal conversion to CHP plants boosts energy system efficiency, lowers operating costs, and minimizes environmental impact.

Using the heat recovered from the solar-thermochemical system and a pyrolysis system, which utilize biomass and petroleum cokes to manage those sources for an environmentally benign operation, ref. [27] proposes a multigeneration plant to produce hydrogen, heat, and power, as well as to achieve product drying. The AspenPlus simulations showed that at about 1300 °C, the optimal $H_2/C$ ratio for an adiabatic hydro-gasifier is 1.5, which maximizes the generation of synthetic methane.

Corrosion tests on copper, aluminum, stainless steel 316, and carbon steel as PCM encasing materials have been analyzed in [28]. Both heating and cooling settings were evaluated.

The authors of [29] improved vacuum-impregnation-based solar-to-thermal power conversion by creating a form-stabilized composite PCM using PEG and porous carbon generated from eggplants.

Dewaxed cotton with 95% cellulose was utilized to create porous carbon with Mg $(OH)_2$ for a shape-stabilized PCM [30]. This porous carbon has a huge surface area due to interconnected holes.

## 3. Proposed Methodology

*Proposed System*

The growing demand for energy, the limited availability of fossil fuels, and the worries over health are the primary factors that are driving the need to transition to renewable energy sources that have significant storage capacity. At this time, our primary emphasis is on the development of CGTs that make use of thermal horsepower and low-pressure steam turbines. The testing was carried out using a solar thermal power plant that had a capacity of 3.5 megawatts and 16 h of thermal storage. The efficiency of a combined heat and power (CCGT) plant may decrease by eight points, LHV, if a capture plant is added to the system. This decrease is contingent upon the position of the steam extraction. Monoethanolamine (MEA) solutions that are based on water are used by the majority of post-combustion capture systems for the purpose of amine-based absorption. We conducted an analysis of the effectiveness of a solar thermal energy facility that has a capacity of 3.5 kWe (1 kWh) and thermal storage that is available for 24 h. The utilization of composite heat battery packs that make use of concentrated solar power (CSP) is becoming an increasingly popular trend in the pursuit of satisfying energy needs while simultaneously reducing the effect on the environment. The approach that was used in this study is shown in Figure 1, and Table 1, For the goal of conducting experiments, two different systems are built. Research conducted on several technologies for the storage of energy indicates that compressed air, driven hydro, and beneath thermal storage systems are the most efficient options for large-scale energy storage. Because of this, superconductors have very low losses in their ability to store energy. In comparison to cars that are powered by petrol, fuel cells are better due to their fuel efficiency. Flywheels are best suited for little effort because of their limited

adjustability with regard to different situations. Salt and stone that have been melted are used as fuel by plants that create heat from the sun.

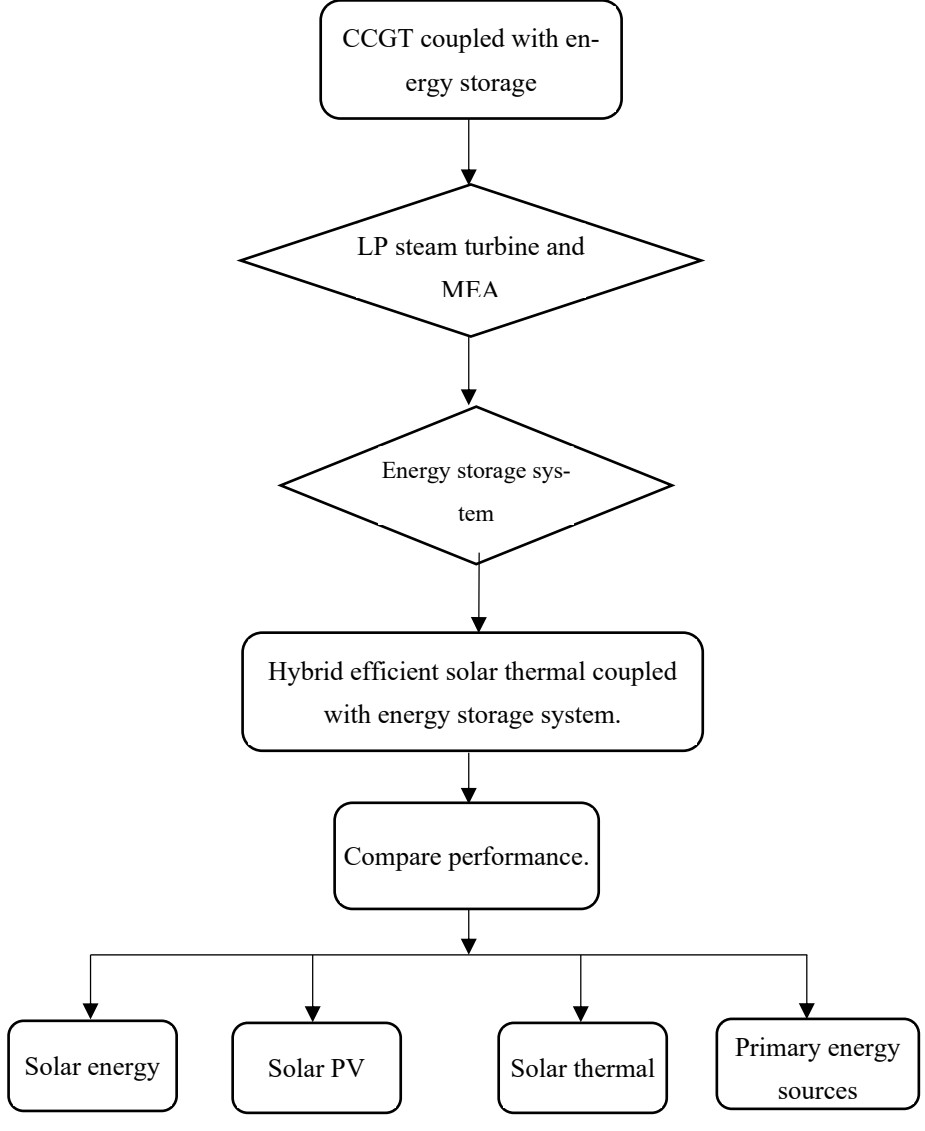

**Figure 1.** Proposed System Model.

**Table 1.** Equipment list used in the experimental setup.

| Name | Utilization Rate (%) | Maintenance Breakdown |
| --- | --- | --- |
| Unit for Solar Tracking | 100 | Replacement Control System Parts |
| System for Heat Transmission (Moving, Heater Feeding Pump, MS/Copper Core) | 75 | Improved steam generation was achieved by revising piping work and the Core's Cast-Iron crucible design. |
| Essential Components from RPI, USA | 75 | The size of the Crystal revised |
| Optical Fresnel | 100 | - |

Figure 2 shows a simplified representation of the basic CCGT designs. The process of exothermic conversion is what heat engines go through to generate heat. This process results in the production of mechanized labor. Fuel is burned in the combustion chamber, producing heat, which is essential for the engine to function properly. Engines may be classified into two categories: internal combustion engines and external combustion engines.

Both of these categories fall within the category of combustion technology. The combustor and the oxidizer, often air, are the two components used by an internal combustion engine (IC engine) to accomplish the task of burning fuel. A functional fluid flow circuit is not complete without a combustor as one of its vital components. Working fluids in combustion machines use thermodynamic cycling to complete the tasks they are responsible for. After going through this moveable, equilibrium, and iso-baric cycle, the fluid will finally return to the state brought about by its thermodynamic qualities. There is a cycle that is often used in gas turbines, and it is called the Brayton cycle. When an internal combustion engine is equipped with a combustor situated outside of the engine, it is feasible for the working fluid to be heated.

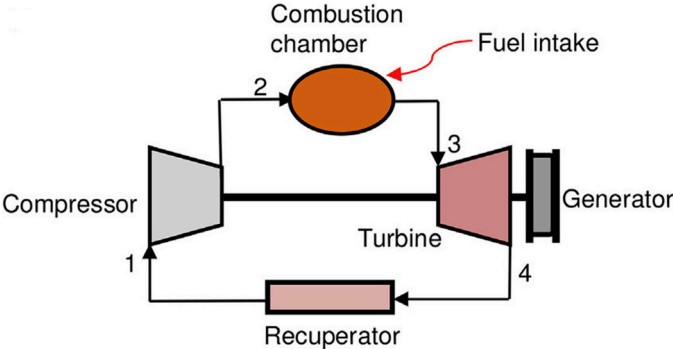

**Figure 2.** Gas cycle for Turbine system.

A heat exchanger transports the heat to the working fluid. Two examples of EC cycles are the Rankine cycle and the Stirling cycle. More and more people are turning to solar gas turbines (SGTs) to produce power. Efficiency and power production improvement is achieved by combining solar energy with the Brayton cycle in this turbine. A gas turbine (GT) is an internal combustion engine (IC) traditionally using air as its operating fluid. In its most basic form, the process begins with introducing air into a compressor, which is pressurized before being directed into a combustion chamber. The next step is to mix the fuel and compressed air, and then the combustion chamber is used to burn the mixture. However, the expansion (exhaust) happens in the turbine part, which is how mechanical power is generated from the expanding air. The GT maintains the working fluid's gaseous phase throughout the cycle. A total thermal efficiency boost of around 60% is achieved by driving a steam turbine using the exhaust heat. Worries about global warming and energy insecurity have accelerated the use of renewable energy sources.

The three main types of CCGT technology that may be employed are post-combustion, pre-combustion, and oxyfuel combustion capture. Carbon capture facilities developed for commercial use and their global capture capacity are shown graphically below. Currently, the oxyfuel combustion capture technique is not being used by any operational full-scale CCGT plant. There has to be some thought given to this. A "pre-combustion carbon capture" system removes carbon dioxide gas from biomass or fossil fuels before they are burned. The overall layout of its configuration is shown in Figure 3. Power plants that employ natural gas or gasification with coal, gas, or biomass make the most common use of it [31]. Partly, oxidation gasification is the first phase in a gasification power plant's standard pre-combustion carbon capture system. The result is syngas, or synthesis gas rich in hydrogen and carbon monoxide. Carbon monoxide and steam react in a water gas shift reformer (WGS) to create $CO_2$ and H+. The waste stream at an industrial plant that focuses on desulfurization and carbon dioxide removal is where the reaction occurs. When the cyclone separation machine finishes cleaning the syngas, we will consider this step finished. Fuel cells, gas turbines, gas boilers, and other power-generating devices that discharge a tiny quantity of sulfur dioxide may use hydrogen petrol, which this carbon capture method produces. This process also uses hydrogen, which is essential for making hydrogen fuel. Because of this, the value of gasoline increases when carbon levels decrease.

$CO_2$ separation is typically performed in natural gas power plants after WGS operations and auto thermal reforming (steam reforming) are finished. Pre-combustion capture is a common term for streams produced by operations with greater pressures, carbon dioxide concentrations, and temperatures between 200 and 400 degrees Celsius. These streams are a result of the same processes that created them. Results from the catalyzed WGS process typically show a hydrogen content of 64–73 mole percent and a carbon dioxide content of 20–23 mole percent in the syngas stream. Compared to the post-combustion capture method, the thermodynamic driving force behind carbon dioxide adsorption is the high partial pressure of $CO_2$. This has decreased energy consumption for the carbon capture and compression operation. The majority of greenhouse gas emissions come from carbon dioxide. The removal and storage of carbon dioxide from the atmosphere is known as carbon sequestration. In an effort to slow the rate of climate change, it is one strategy for lowering atmospheric carbon dioxide levels.

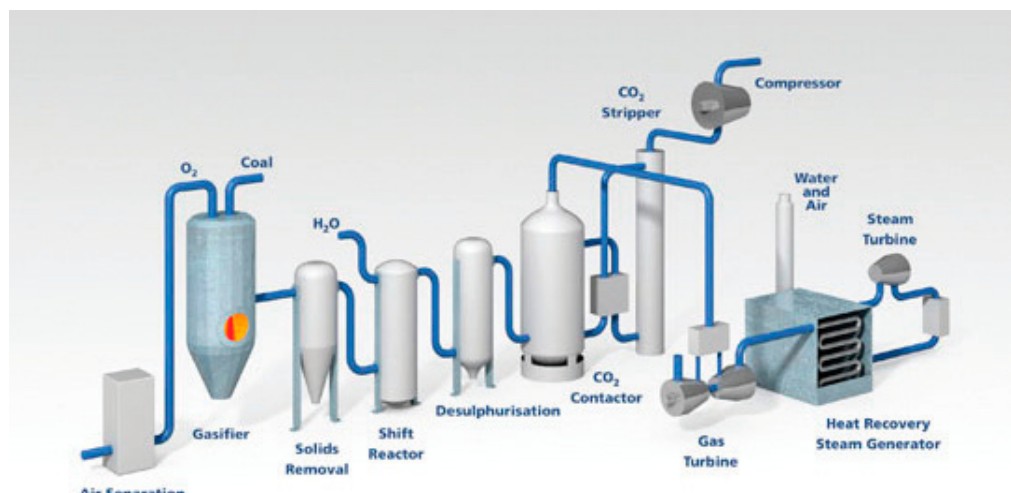

**Figure 3.** Carbon sequestration system.

Figure 4 is a schematic depicting a system of solar panels. One can see every component, from the charge controller and inverter to the batteries and the DC and AC load, in this photo. These are the tools that have been used during the trial. With photovoltaic technology, consumers may generate electricity in an eco-friendly, silent, and reliable way. Hundreds of individual solar cells are housed in a waterproof and sealed module. The goal is to make the most of solar thermal systems. Both series and parallel linking of modules are feasible. The versatility of the modular solar thermal system allows designers to create solar power systems that can meet various electrical needs. A solar thermal system often falls into one of two categories: one system is "grid-tied," which is connected to the utility grid; the other is standalone, disconnected from the public electrical grid. Since grid-tied solar systems are permanently attached to the power grid, battery storage is unnecessary. Reduced utility power use is possible with the help of a solar thermal system, which may generate enough energy to power a home or business. The PV system is designed to automatically feed any extra electricity into the grid if its production exceeds the power needs. A solar thermal system cannot work without a battery backup during a power outage. Standalone systems are independent and do not need any external power source as they are not linked to the utility grid. During the day, they generate energy; at night, they store excess power. Although there are several parts to a PV system, the ones that generate electricity—the PV cells—are the most crucial. However, more components are needed to control, convert, and store the energy. This system includes batteries, charge controllers, inverters, and solar thermal modules.

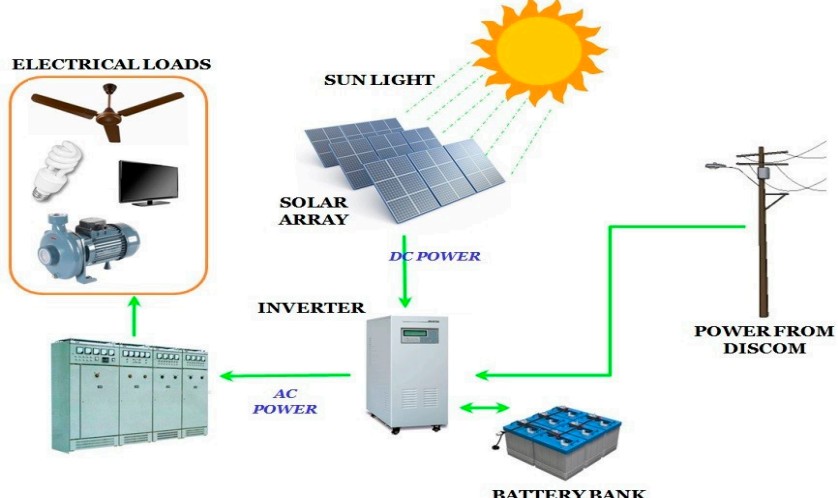

**Figure 4.** Solar Energy System.

- Energy-generating solar thermal modules:

The module's voltage defines the model's number of cells in solar thermal modules. By connecting solar thermal cells in series or parallel, arrays of these cells have the potential to generate more electricity than a single cell could on its own. In most cases, the system's nominal voltage must be identical to the nominal voltage of the storage system. The efficiency and cost-effectiveness of solar modules are the primary characteristics considered while evaluating these modules. Crystalline and thin film materials are the two primary components used to manufacture solar modules. Both polycrystalline and monocrystalline modules make up the majority of crystalline semiconductors. A monocrystalline module is composed of a single and enormous silicon crystal alone. They are the most efficient at turning sunlight into energy, with an efficiency of over 18%. The price is also more than that of polycrystalline varieties. Solar thermal modules may be either polycrystalline or monocrystalline. Even though they are not as efficient as monocrystalline modules, polycrystalline ones are much more common in PV systems. Efficiency ratings for polycrystalline modules are between 13% and 15%.

- Batteries and charge controllers:

The batteries' primary function is to store electrical energy for later use, even when sunlight is not available. Two main types of storage batteries are used much more often in solar thermal systems. Each of these two categories of batteries is referred to by their respective names: deep-cycle batteries and lead-acid batteries. In addition to being the most common battery, lead-acid batteries can deliver very high currents. Lead-acid batteries power the majority of electronic gadgets. Lead-acid batteries are advantageous for several reasons, including the fact that they are inexpensive and widely available. A charge controller is a piece of equipment that establishes a connection between the solar array's output and the energy stored in the batteries. This component is designed to control and manage the flow of the current throughout the system. To provide a more in-depth explanation, these systems are intended to prevent the batteries from being charged or drained over their capacity. Certain charge controllers make use of pulse width modulation as a method of controlling whether the device is on or off. It is necessary to carry out this action to accomplish the desired result. It does this by setting the battery to its total capacity and progressively reducing the amount of charge added to it as it approaches full charge.

Electric devices or the grid may obtain their power from solar panels or batteries thanks to inverters, which change the voltage from direct (DC) to alternating (AC). Another use for inverters is transforming the AC power generated by solar panels. One kind of inverter is the standalone kind. These inverters cannot connect to the grid, but they power the system's electrical equipment. It is designed for the grid-tie inverters to disconnect

and turn off promptly during a utility supply outage, but they will not keep the lights on during a blackout. Connecting grid-tie inverters to the grid is something they can do automatically. It is a hybrid technology combining the two types of battery backup inverters that were previously available. They can be connected to the grid and provide the necessary electricity to the system.

## 4. Experimental Setup

The growth of green energy sources and the assurance of power security are both viable answers to the environmental issues that we face. A great deal of heat and electrical energy may be produced by solar panels. This chapter explains the CCGT-energy storage system technique. Solar thermal system sales could be boosted with an energy storage system. The first setup for the project is a "1 MW electrical (3.5 MW thermal) renewable energy facility with 16 h of heating potential." The planned 1 MW solar thermal power plant uses Parabolic Solar Reflectors to convert solar energy into electricity at a 12% efficiency, and it has 16 h of storage capacity. The second trial is a thermal energy storage system with a high energy density for a concentrated solar power plant. The parabolic solar reflector is 60 square meters in area. It utilizes an articulated mirror space frame made of solar-grade material. An external frame with cross and long bars, a central bar gap, a solid base, and a wheel in motion make up the structure. The PSR is constructed entirely from mild steel that meets the requirements of ISO 2062 for its solid parts and ISO 4923 for its hollow parts. We apply a thick coat of epoxy and polyurea to all exposed surfaces to ensure their durability and lifespan.

The following expression describes the process by which thermal energy (perceivable heat) is stored in a solid mass by increasing its temperature:

$$Q = V \, C_p DT$$

where (Q) is the quantity of thermal energy (J) stored in the material, (V) is its volume ($m^3$), and (DT) is its density ($kg/m^3$).

The specific heat of a material is denoted by the formula $C_p = J/(kg \, °C)$.

Difference in Temperature (°C)

The temperature at which 1 cubic meter of salt would store 300 kilowatt-hours of energy (300 kilowatt-hours per cubic meter, or $KWh/m^3$) is given as follows:

$$DT = Q / \left( V \, C_p \right)$$

$$KWh = KJ \, 3600 \text{ (seconds per hour)}$$

$$\text{Quantum energy (Q)} = 3600 \times 300 = 1.08 \times 10^6 \text{ KJ}$$

$$Q = 1.08 \times 10^9 \text{ J } V = 1 \text{ m}^3$$

$$\rho = 2200 \text{ kg/m}^3$$

The critical temperature, $C_p$, is 870 J/kg °C.
Based on these parameters, we obtain the following:

$$DT = 565 \, °C$$

As a result, a core temperature of 565 °C is required to harvest and store 300 $KWh/m^3$ of thermal energy density.

### 4.1. Analysis and Interpretation

All models will be simulated and analyzed on a 64-bit Windows 11 system with an Intel I5 CPU and 8 GB of RAM. We will test and evaluate the proposed and existing models in various WSN network configurations.

*Average Delay*: Average time a packet takes to travel from origin to destination. The formula is as follows:

$$D = \frac{\sum_{i=1}^{N} d_t^i + d_p^i + d_{pc}^i + d_q^i}{N} \tag{1}$$

where $N$ is the total number of transmission links, $d_t^i$ is the transmission delay of $i^{th}$ link, $d_p^i$ is the propagation delay, $d_{pc}^i$ is the processing delay, and $d_q^i$ is the transmission delay.

*Average Throughput*: This metric counts packets per second. The usual Kbps transfer rate is as follows:

$$T = \left( \frac{R}{T^2 - T^1} \right) \times \left( \frac{8}{1000} \right) \tag{2}$$

where $R$ is the complete received packets at all destination nodes, $T^2$ is the simulation stop time, and $T^1$ is the simulation start time.

*Average Energy Consumption:* The network average is calculated by adding together each node's energy usage after the simulation. The total energy consumption formula, $E^{tot}$, is as follows:

$$E^{tot} = \sum_{i=1}^{N} E_i^{initial} - E_i^{consumed} \tag{3}$$

where $E_i^{initial}$ and $E_i^{consumed}$ are the initial and consumed energy of $i^{th}$ node, respectively. $N$ is the total number of nodes in the network. The average consumed energy is computed as follows:

$$E^{avg} = \frac{E^{tot}}{N} \tag{4}$$

E: Network Lifetime
Lifetime (Rounds) = Total Remaining Energy/10
*PDR*: The ratio of packets transmitted and received is calculated. The formula is as follows:

$$P = \left( \frac{P_r}{P_g} \right) \times 100 \tag{5}$$

where, $P_r$ is the number of received packets, and $P_g$ is the number of generated packets.

*Communication Overhead*: Number of routing packets compared to network data packets. The formula is as follows:

$$O = \sum_t \left( \frac{RT^t}{DT^t} \right) \tag{6}$$

where, $RT^t$ is the total number of routing packets, and $DT^t$ is the total number of data packets at time $t$.

### 4.2. Simulation and Results

A combined cycle power plant (CCGT) combined with a storage system is an exciting development in the quest for constant power. The experimental outcomes of both setups compare well. Based on the findings of the experiments, it is clear that solid-state storage devices are the best choice. Only now can we give such a comprehensive analysis and comparison of situations. Figure 5 shows the proportionate contribution to the module's total radiation, thermal loss, and thermal gain from each individual receiver. Unlike PV/T systems, conventional solar thermal collectors may be built with thermal loss-reducing features, like double-glazing and black selective absorbers, and they do not rely on PV cells, which lose efficiency at higher temperatures. High operating temperatures render PV/T collectors inefficient. A heat exchanger (HE) located at the storage tank's base allows the heat from a PV/T collector's HRF to be transferred to the water inside. This configuration is used in flat-plate thermosiphonic units (FPTUs) and similar systems. In contrast, the FPTU system's HRF uses an appropriately installed HE to move heat from the collectors

to the tank's central and upper storage areas. A comparable system may be produced by substituting evacuated tubes for flat-plate collectors.

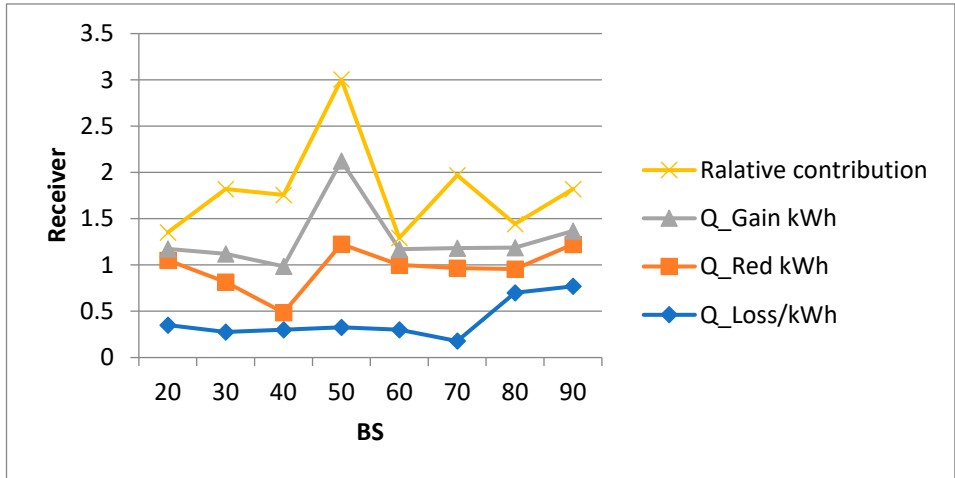

**Figure 5.** Proportionate contribution to the module's total radiation, thermal loss, and thermal gain.

The area of solar radiation that a surface receives per unit area when held perpendicular to the rays that arrive in a straight line from the sun's direction at its present location in the sky is called Direct Normal Irradiation (DNI). The daily irradiation in $Wh/m^2$ is calculated when all the hourly readings are added together. Assuming a constant irradiance of $100 \ W/m^2$ for 10 h, the total daily irradiation would be $1000 \ Wh/m^2$, or $1 \ kWh/m^2$. For 10 min values, the process is the same; however, after adding up all of the data, divide the total by 6. Every 10 min, the location's actual Direct Normal Irradiance (DNI) is determined and then averaged across an hour and a day. Therefore, the output of the Reflector = aperture area (SQM) × avg. DNI for the day (kWhrs) × efficiency factor. Figure 6 displays the average domestic product of the area in 2010.

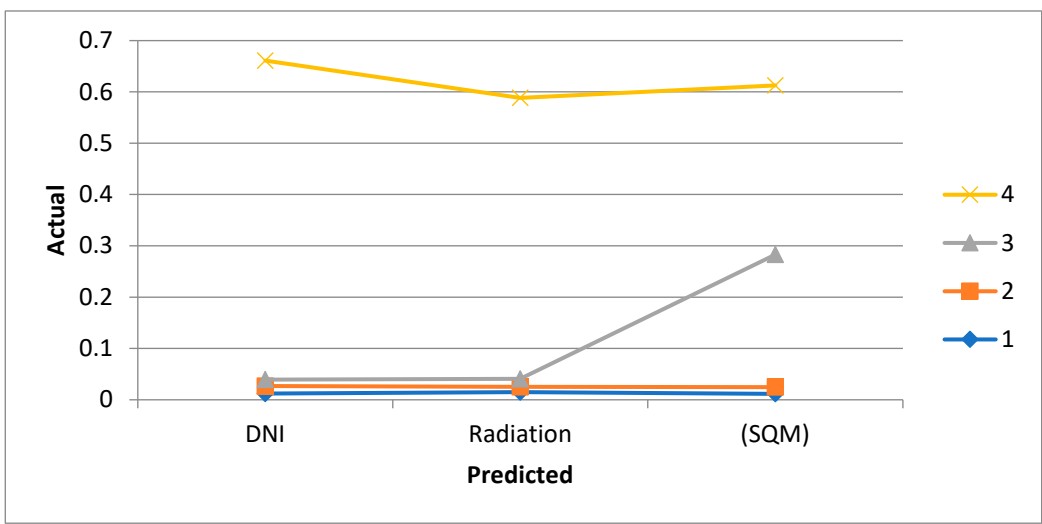

**Figure 6.** DNI Radiation in kWhrs/day from avg.

Figure 7 displays the thermal output of 60 SQM reflectors. The most common kind of reflector used in solar thermal electric applications is glass mirrors. In solar concentrators, Fresnel lenses bend and concentrate light rays from a broad region to shine on a specified target. Based on the intended function, the concentrator's construction material changes. Mirrors are the material of choice for solar thermal concentrators, while glass or clear plastic are acceptable options for BIPV systems. Compared to PV materials, these materials are much less expensive. The chemical properties and vast surface area of porous zeolite

allow it to absorb significant amounts of nitrogen under high pressure. Air is compressed and passed over the zeolite in the oxygen concentrator, which causes the zeolite to absorb nitrogen from the air.

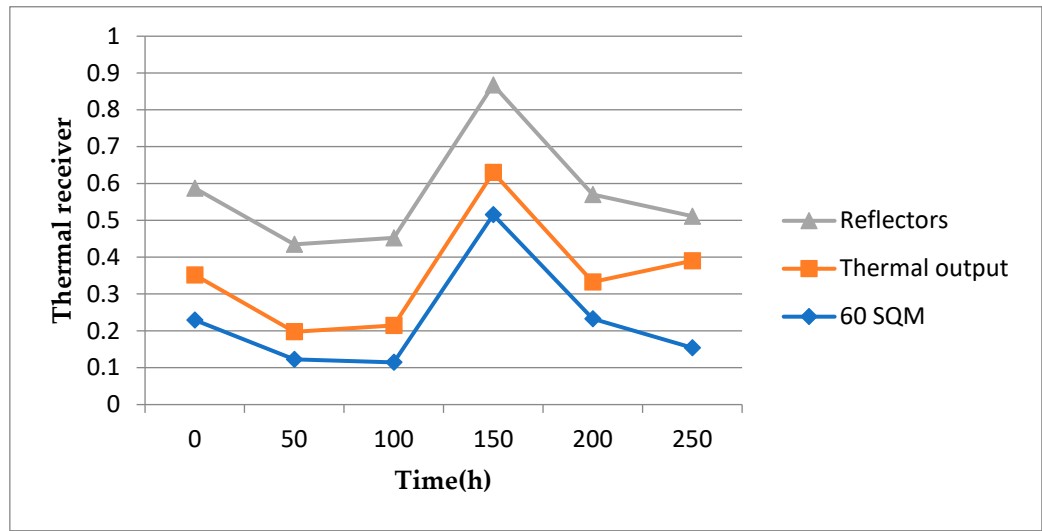

**Figure 7.** Thermal output of 60 SQM reflectors.

The Static receiver obtains its charge throughout the day from the sun rays that are focused by the parabolic reflector, which can be seen in Figure 8. A charge is applied to the receiver in the absence of the front glass cover. The highest temperature that was measured was 400 degrees Celsius. Glasses with varying levels of lead content have thermal conductivities at room temperature ranging from 1.39 W/(m·K) for pure quartz glass to around 0.51 W/(m·K). The range of values for the most popular silicate glasses is from 0.92 to 1.21 W/(m·K). Air has a thermal conductivity of only 0.028, but glass's thermal conductivity is 0.687, a 27-fold increase. The typical figure for a plain glass window is 6.0 W/m$^2$K, whereas for a double-glazed low-emission window, it is 1.0 W/m$^2$K. Heat moving from one surface to another involves every single particle on that surface. More surface particles can transfer heat from a more oversized item. A one-to-one relationship exists between the surface area and the heat transmission rate.

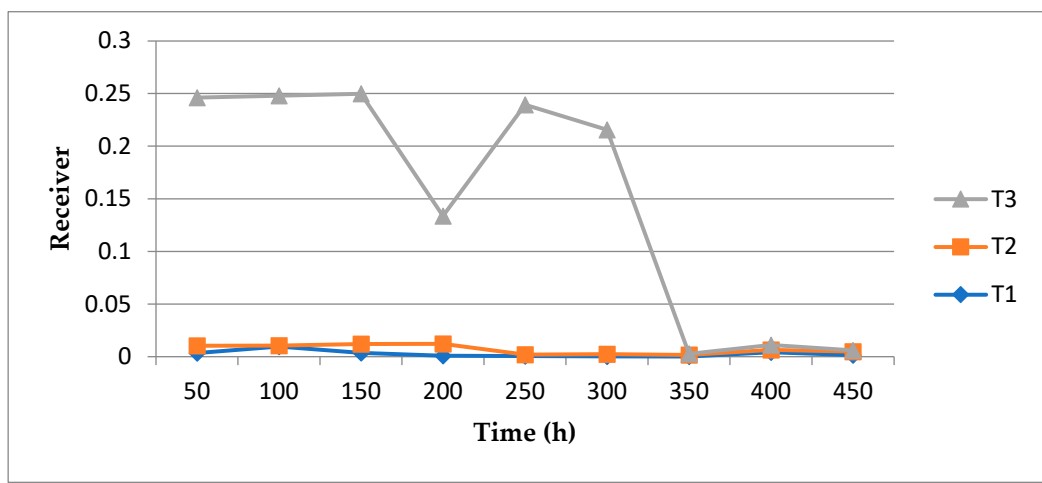

**Figure 8.** Thermal behavior of the receiver without front glass.

## 5. Conclusions

Priorities on a global scale include energy security, high efficiency, environmental protection, sustainable development, and economic viability. One of the key factors that has

been driving the growing rise in energy consumption that has happened over the course of the previous few decades is the ever-increasing demand that people have for modern conveniences and pleasures. As a consequence of this, it is of the highest significance to design energy solutions that are not only environmentally friendly but also operate continuously across the board. Typical renewable power systems have a number of important obstacles, the most critical of which are the following: ensuring a constant supply of electricity, storing that energy for later use, and reducing the emission of greenhouse gases. The standard model for renewable energy, which uses a passive hybrid approach, might be replaced with an alternative that utilizes energy storage in conjunction with a combined cycle power plant (CCGT) system. This would be an alternative to the traditional model. In light of the fact that the hybrid system is able to harness the power of the sun, it is capable of producing a substantial amount of energy. There are a variety of benefits that are linked with these systems, some of which include low costs of operation and maintenance, acceptable efficiency, and the lack of greenhouse gas emissions while they are in operation. As a way of creating steam for the carbon capture and sequestration system, our recommendation is to continue using solar thermal storage as a means of producing steam. Solar thermal energy is the most widespread of all renewable energy sources. It is also the most ubiquitous. When it comes to heating swimming pools, heating water for household use, and building interior spaces, the material is often used for these aforementioned purposes.

The solar savings fraction, the percentage of overall energy usage that can be attributed to solar technology, is an essential metric to keep in mind while talking about solar power. By combining active solar technology with passive solar systems, traditional energy savings might be far larger than when utilizing passive solar systems alone to heat a space. A number of devices exist that have the potential to transform solar radiation into usable electricity; solar ponds and towers are only two examples. Despite this, the solar stirling engine achieves an astounding 40% efficiency. When geothermal power stations release carbon dioxide into the atmosphere, it is entirely because of natural processes. Solar thermal systems have energy and waste management challenges due to the processing of raw materials. In addition, solar thermal systems alone will not be enough to achieve future sustainable development targets. The greatest possible method for mitigating climate change is the combination of EGSs with granites, which produce a great deal of heat.

**Author Contributions:** Y.A.H.A.-E.: Conceptualization, Data curation, Formal analysis, Investigation, Methodology, Project administration, funding acquisition, Resources, Software, Validation, Visualization, Writing—review & editing, Writing—original draft; M.Y.: Conceptualization, Funding acquisition, Project administration, Resources, Supervision, Validation. All authors have read and agreed to the published version of the manuscript.

**Funding:** This study is supported by FEN-B-101013-0399 and FEN-D-091116-0505 at Marmara University.

**Institutional Review Board Statement:** Not applicable.

**Informed Consent Statement:** Not applicable.

**Data Availability Statement:** The data presented in this study are available on request from the corresponding author.

**Conflicts of Interest:** The authors declare no conflict of interest.

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
