# Peer review of "Integrated Systems of a Solar Thermal Energy Driven Power Plant"

_applsci, doi:10.3390/app14052088_

Round 1
Reviewer 1 Report
Comments and Suggestions for Authors
Hammady et al. reported a user-friendly Visual Basic simulation program for mathematical modeling of the solar thermal plant's thermal performance. The created simulator may help to improve the energy efficiency of the power plants. This work can be accepted after the following issues are addressed.
1. Some number should be subscripted, such as “MgCl2.6H2O and CaCl2.6H2O”. Please check the full text.
2. In the “Proposed system” section, Figure 2-4 are not mentioned in the text. In addition, there are two “Figure 2”, two “Figure 3”, and two “Figure 4”, please check.
3. In Figure 4 of the “Simulation and Resul” section, the legend for x and y axis should be added. In addition, is the “resul” means “result”?
4. In the first sentence of the introduction, some related references should be cited, such as DeCarbon, 2024, 3, 100033; DeCarbon, 2024, 3, 100025; J. Semicond, 2022, 43(12): 122701.
5. In Figure 4 of the “Proposed system” section, the electricity comes from photovoltaic panel, the main content of this paper is solar thermal system, better use solar thermal system instead of the solar panel.
Comments on the Quality of English LanguageQuality of english language is high.
Author Response
Dear Reviewer,
Thank you for evaluating our paper, we have updated last version of the manuscript with the response letter, please find the attachment.
Kind Regards

Reviewer 2 Report
Comments and Suggestions for Authors
Very poor research and manuscript.
A major revision, in fact a complete re-structuring, updating, and scientific approach are needed.

No specific comments, but to be improved (also for numerous typos.
Author Response
Dear Reviewer,
Thank you for your evaluating our paper.
Please find the response letter attached.
Kind Regards

Round 2
Reviewer 2 Report
Comments and Suggestions for Authors
Thanks for including answers to the initial comments.
Although some suggested extra references were not included in the revision; I accept the revised manuscript.
No further comments
Comments on the Quality of English LanguageMinor language imperfections still need to be corrected.